# Next Generation CD44v6-Specific CAR-NK Cells Effective against Triple Negative Breast Cancer

**DOI:** 10.3390/ijms24109038

**Published:** 2023-05-20

**Authors:** Martin J. Raftery, Alexander Sebastian Franzén, Clarissa Radecke, Abdelhadi Boulifa, Günther Schönrich, Sebastian Stintzing, Jens-Uwe Blohmer, Gabriele Pecher

**Affiliations:** 1Berlin Institute of Health at Charité – Universitätsmedizin Berlin, Charitéplatz 1, 10117 Berlin, Germany; 2Onkologie und Tumorimmunologie, CCM, Charité – Universitätsmedizin Berlin, Corporate Member of Freie Universität Berlin and Humboldt-Universität zu Berlin, Medizinische Klinik m. S. Hämatologie, Charitéplatz 1, 10117 Berlin, Germany; 3Institute of Virology, CCM, Charité – Universitätsmedizin Berlin, Corporate Member of Freie Universität Berlin and Humboldt-Universität zu Berlin, Charitéplatz 1, 10117 Berlin, Germany; 4Department of Gynecology and Breast Cancer Center, Charité – Universitätsmedizin Berlin, Corporate Member of Freie Universität Berlin and Humboldt-Universität zu Berlin, Charitéplatz 1, 10117 Berlin, Germany

**Keywords:** cancer, immunotherapy, CAR, CAR-NK, TNBC, CD44v6

## Abstract

There is a medical need to develop new and effective therapies against triple-negative breast cancer (TNBC). Chimeric antigen receptor (CAR) natural killer (NK) cells are a promising alternative to CAR-T cell therapy for cancer. A search for a suitable target in TNBC identified CD44v6, an adhesion molecule expressed in lymphomas, leukemias and solid tumors that is implicated in tumorigenesis and metastases. We have developed a next-generation CAR targeting CD44v6 that incorporates IL-15 superagonist and checkpoint inhibitor molecules. We could show that CD44v6 CAR-NK cells demonstrated effective cytotoxicity against TNBC in 3D spheroid models. The IL-15 superagonist was specifically released upon recognition of CD44v6 on TNBC and contributed to the cytotoxic attack. PD1 ligands are upregulated in TNBC and contribute to the immunosuppressive tumor microenvironment (TME). Competitive inhibition of PD1 neutralized inhibition by PD1 ligands expressed on TNBC. In total, CD44v6 CAR-NK cells are resistant to TME immunosuppression and offer a new therapeutic option for the treatment of BC, including TNBC.

## 1. Introduction

Current therapy options for breast cancer (BC) are relatively effective in the subsets of cancers where hormone receptors are expressed. In the remaining subsets of BC where these receptors are missing, such as triple-negative breast cancer (TNBC), the prognosis is much worse. TNBC is a diverse group of tumors that occur most frequently in younger women [1] and are clinically more aggressive. Metastasis of TNBC is often lethal [2], and therapeutic options for these patients are very limited.

Cellular therapy using chimeric antigen receptors (CAR) is now an established treatment for hematological tumors, with FDA approval for CD19 or B-cell maturation antigen (BCMA)-positive tumors [3,4]. The expression of a synthetic targeting receptor or CAR in an immune cell with cytotoxic capacity redirects cellular attack to the target of the CAR. The process involves extracting cells from the patient, creating the cellular product by expression of the receptor and then reintroducing the cells to the patient. Conventionally, T cells have been used for this personalized therapy. The receptor-expressing cells can remain in the patient for very long periods of time, and clinical response is related to long-term engraftment and persistence [5,6,7,8]. Despite its effectiveness, the process is limited to autologous cells, due to the risk of side effects such as graft versus host disease (GvHD). A further problem is that only some tumors are susceptible to this therapy, and solid tumors in particular are frequently resistant to CAR treatment. This is in part because solid tumors generate a tumor microenvironment (TME) that is immunosuppressive, thus protecting them from effective immune attack. In order to overcome these problems, many groups have started to work with natural killer (NK) cells, which offer a highly cytotoxic alternative to T cells with a different sensitivity to the TME. Current expansion and cultivation protocols, combined with recent developments in gene transfer, allow the facile generation of CAR-transduced NK cells and the development of an allogenic cellular product [9].

Here, we present a novel CAR specific for CD44v6, an antigen present in the majority of BC. CD44 is a highly spliced cell-surface glycoprotein with multiple roles in adhesion and proliferation. CD44 and its subvariants have been associated with poor prognosis in cancer patients and the splice variant CD44v6 has been implicated in tumorigenesis, tumor cell invasion and metastasis [10]. Our CD44v6 CAR vector expressed in NK cells is designed to overcome the tumor microenvironment, engraft and effectively treat solid BC tumors.

## 2. Results

We generated a novel CD44v6-specific lentivirus vector (Figure 1) by combining the following elements. Initially, a CD44v6-specific CAR was constructed by fusing CD44v6-specific Vl and Vh recognition domains with the hinge region of IgG1, the transmembrane and signaling domains of CD28 and the signaling domain of CD3ζ (Figure 1). This was coupled by P2A sequences to a checkpoint inhibitor (PD1x) and a suicide gene (thymidine kinase from herpes simplex virus type 1; HSV TK). This was followed by a response element with an IL-15 superagonist (15R15) under the control of an NFAT promoter.

In order to block the PD1 checkpoint molecule, we identified a natural splice variant of human PD1 that lacks the cytoplasmic signaling domain and thus acts as a competitive inhibitor of PD1. We reasoned that a naturally occurring molecule would minimize the risk of off-target interactions or unwanted immune responses. We further modified this molecule by including an A132I mutation to enhance ligand affinity, similar to previously reported mutations [11,12]. In order to allow the elimination of the vector, we included a suicide gene, HSV TK. We modified HSV TK with an A168H mutation, as natural and induced mutations at this amino acid residue decreased affinity to endogenous substrates and increased affinity to ganciclovir (GCV) [13,14,15]. In order to create an IL-15 superagonist, we fused the signal sequence and sushi domain of IL-15Rα via a novel 21aa linker to the IL-15 gene. We also included an N72D mutation in IL-15 in order to enhance binding to the IL-15Rβ chain, as previously reported [16]. The finalized vector was codon optimized, including the removal of potential splice sites and cloned into a lentivirus vector.

We introduced our vector by lentivirus transduction into reporter Jurkat cells [17]. The expression of the CD44v6 CAR was confirmed by flow cytometry and was comparable to expression after transduction of the unrelated gene PD1 (Figure 2A). Similarly, the NK cell line YT or primary NK cells from peripheral blood were transduced with a CD44v6 CAR vector or a control vector, which is identical but lacks the CD44v6 recognition site. Transduced primary NK cells showed expression of both CAR constructs (Figure 2B). The phenotype of transduced NK cells was maintained, with over 90% expressing CD56 and CD94 and less than 1% contaminating CD3 T cells. Approximately half of the NK cells expressed CD62L, indicating homing potential. The cells were not terminally differentiated, as indicated by low CD57 expression, showing replicative potential. Although the NK cells were not terminally differentiated, they had high functionality, as indicated by the expression of NKG2D (Figure 2B).

In order to validate the tumoricidal action of CD44v6 CAR, we investigated BC cell lines, including three TNBC cell lines (MDA-MB-231, MDA-MB-468 and HCC1937). BC cell lines showed significant levels of expression of CD44v6, making them a suitable therapeutic target (Figure 3A). In order to show the functionality of the CD44v6 vector, we utilized a Jurkat reporter cell line that expresses GFP under the control of an NFkB-responsive promoter. These cells were transfected with the CAR44v6 vector or the control vector. Cells transfected with the control vector show similar levels of CAR on their cell surface as the CD44v6 vector, but do not respond specifically to antigen. Reporter cell lines exposed to four BC lines showed specific stimulation (Figure 3B). In total, the CD44v6 CAR recognizes CD44v6 on TNBC lines.

Checkpoint molecule expression is a significant immune evasion mechanism in oncogenesis. Stimulation via the PD1 molecule results in attenuation of the T cell receptor (TCR) signal and blunting of T cell attack. CAR molecules typically use elements such as the CD3ζ signaling domain, which makes them susceptible to this inhibitory mechanism. We first established by flow cytometry that BC lines often express high levels of PD-L1, the most common PD1 ligand (Figure 4A). No significant staining was seen for PD-L2, the second PD1 ligand, in the cell lines tested. In order to evaluate the PD1 signaling in this system, we used a Jurkat reporter cell line expressing high levels of functional PD1 on the cell surface (Figure 2A). After mixing reporter and target cells for 24 h, a comparison between CD44v6 CAR Jurkat cells and a CD44v4 CAR vector where the PD1x molecule was removed (Figure 4B) showed that PD1-expressing CD44v6 CAR Jurkat reporter cells lacking the vector-encoded checkpoint inhibition molecule were less sensitive to stimulation, as expected. In the case of TNBC line HCC1937, which expresses high levels of both CD44v6 (Figure 3A) and PD-L1 (Figure 4A), the inhibition was fourfold. This result was confirmed by preincubating for 1 h both reporter cell lines in the normal medium at 4 °C with a mixture of checkpoint inhibitor monoclonal antibodies pembrolizumab and atezolizumab at a concentration of 10 μg/mL each before adding Jurkat reporters to target cells for 24 h (Figure 4B). In particular, for HCC1937, the inhibitory effect of checkpoint molecules in the absence of vector-encoded checkpoint inhibitors was fully removed by soluble checkpoint inhibitor therapeutics. Taken together, this data shows the functionality of the checkpoint inhibition in our vector.

In order to improve engraftment and expansion of CD44v6 CAR NK cells in vivo, we included an IL-15 superagonist (15R15) in our vector. The supernatant from Hek 293 cells transfected with a plasmid expressing 15R15 under the control of a strong constitutive promoter showed high levels of IL-15 activity (Figure 5A), similar to IL-2 and IL-15. Then, 15R15 was released from Jurkat or primary NK cells transduced with the CD44v6 vector after exposure to different BC lines, whereas cells transduced with the vector lacking the CD44v6 recognition domain released little 15R15 (Figure 5B). Cytokine release could be induced at both low and high effector-target ratios (Figure 5C). Our novel CD44v6 CAR also includes a suicide gene to allow the removal of the CAR-NK cells if required therapeutically. We included a codon-optimized version of the HSV-1 thymidine kinase gene, which converts inactive substrates, such as ganciclovir (GCV), into active intracellular products that disrupt DNA production and induce cellular apoptosis. The addition of different doses of GCV showed the removal of transduced cells (Appendix A). Overall, this data indicates that our next-generation CD44v6 CAR construct shows specific activity against BC lines, including TNBC.

In order to establish the functionality of the CD44v6 CAR as a cellular therapy, we used both the YT cell line and primary NK cells. YT cells are an immortalized NK cell line with cytotoxic potential [18]. YT cells transfected with either the CD44v6 CAR or a control vector where the CD44v6 recognition domain was omitted were used in kill assays against BC lines, showing efficient killing of CD44v6-expressing TNBC cell lines (Figure 6A). This was confirmed using primary NK cells transduced with the CD44v6 CAR vector (Figure 6B). Vectors without checkpoint inhibition (blue line) or 15R15 (red line) performed less well than the full construct. The high cytotoxicity of primary NK cells to tumor cells was reflected in the background killing of the control vector (grey line).

We then tested our novel CD44v6 CAR in a 3D TNBC multicellular tumor spheroid (MCTS) solid tumor model. The addition of fibroblasts to the MCTS allowed us to evaluate the contribution of cancer-associated fibroblasts and the tight stroma that can be seen in BC [19,20]. We first confirmed that CD44v6 is expressed in our 3D model (Figure 7A) and that CD44v6 CAR is effective in 3D models by using CD44v6 CAR-transduced Jurkat reporter cells (Figure 7B).

NK cells transduced with CD44v6 CAR were able to recognize and attack tumor cells despite the presence of some elements of the TME (Figure 7C). To show the validity of the MCTS model, we could demonstrate an even dispersion of HCC1937 cancer cells and MRC5 fibroblasts (Appendix A). Expression of PD-L1 as well as the target molecule CD44v6 was not inhibited by the formation of MCTS (Appendix A). Interestingly, the coculture spheroid model forms smaller, compacter spheroids compared to the cancer cell spheroids alone (Appendix A), which mimic the tight stroma seen in BC TME. Both MCTS models continuously formed uniform spheroids with a rough size of 500 µm in diameter. This is the size at which MCTS has been seen forming a necrotic core, a hypoxic environment, nutrient- and pH-gradient, all of which are known as hypoxic tumor immunoinhibitory characteristics [21,22,23].

## 3. Discussion

Here, we describe a new next-generation CAR specific for CD44v6 with strong activity against TNBC. There are over 70,000 newly diagnosed patients with breast cancer per year in Germany, and the median survival (stage IV) with standard therapy (surgery, chemotherapy and radiation) is only 3 years [24]. There is thus an urgent need for new and more effective therapies.

CD44v6 is a prominent marker on many solid tumors, such as breast, bladder, gastrointestinal, lung and ovarian cancers [25] and has been described on other tumors, such as prostate cancer. Primary and metastatic BC have been shown to express high levels of CD44v6 [25,26], which has been shown to be associated with the invasiveness of BC [27]. A small subset of the main tumor mass termed cancer-initiating cells (CIC) or cancer stem cells (CSC) is thought to be responsible for the capacity of a tumor to metastasize, and these cells typically also express CD44v6 [28]. CD44v6 has been shown to help maintain the stemness of CICs [29]. For these reasons, CD44v6 is thought to contribute to metastasis and therapy failure.

CD44v6 is one of many splice forms of the CD44 hyaluronic acid receptor, an adhesion molecule that consists of an N-terminal globular domain followed by a highly glycosylated stalk domain, transmembrane and cytoplasmic domains. CD44 is widely expressed, but CD44 with variable domains is often expressed only in subpopulations of cells. Variable domains are typically inserted in the stalk domain. This domain is responsible for oligomerization and thus variation can have a powerful effect on the function of CD44. In particular, the interaction with receptor tyrosine kinases, antiapoptotic proteins, G-protein-coupled receptors, membrane-bound proteases and ATP-binding cassette transporters is relevant for cellular transformation. It has been reported that CD44v6 released on exosomes can promote tumor progression [28], presumably by promoting a pro-oncogenic TME.

Although CD44v6 is also expressed to a limited extent on some normal adult cell types, such as keratinocytes and monocytes, CD44v6 CAR T cells have previously been shown to have minimal toxicity to normal keratinocytes and hematopoietic stem cells, although a reversible monocytopenia was observed [29]. In order to avoid GvHD, T cell therapy must be personalized, as CAR-T cells often cause cytokine release syndrome (CRS) or immune effector cell-associated neurotoxicity syndrome (ICANS) [30]. CAR-NK cells only very rarely cause CRS or ICANS [31].

The design of our CAR did not include any NK-specific signaling elements, as prior work has shown that CD28-based CAR-NK cells are therapeutically effective [31]. We also do not anticipate any toxicity problems with our vector for several reasons. Firstly, NK cell therapy induces low levels of toxicity compared to T cell therapies in general. Secondly, although our vector includes a CD28 costimulation and transmembrane domain that has been implicated in T cell toxicity [32] due to endogenous CD28 recruitment [33], this is irrelevant in NK cells, which do not express CD28.

Prior work has indicated that CD44v6 could be a good target for solid tumors. T cells expressing a second-generation CD44v6 CAR have been shown to be effective against lung and ovarian adenocarcinomas in mice [34] and against head and neck squamous cell carcinoma [35], urothelial carcinoma [36] and pancreatic adenocarcinoma [37]. The NK cell line NK-92 carrying a third-generation pan-CD44 CAR also showed cytotoxic activity against ovarian cancer in vitro [38]. Along this line, we demonstrated cytotoxic function in both 2D and 3D models of triple-negative breast cancer. Further modifications, such as the inclusion of chemotaxis receptors, are expected to enhance migration and contribute to overcoming the TME [39].

An accumulating body of evidence indicates that tumor cell cultures in a 3D arrangement are truer to their in vivo counterparts than standard 2D cultures [40,41,42,43]. A tight stroma is known as a negative prognostic marker in breast cancer, and it is associated with poor overall survival [19,44]. One explanation for this is that a tight stroma creates a physical barrier, which makes it harder for antitumorigenic immune cells, such as NK cells, to penetrate and reach the tumor cells [43,45]. Furthermore, it has been shown that breast cancers have depositions of collagen and a stiff ECM, which can be associated with more aggressive subtypes of breast cancer [20]. Fibroblasts are the main ECM-producing cell type [44,45,46] and therefore are believed to be the culprits behind the tight stroma seen in breast cancer. Our CD44v6 CAR-NK showed activity against both 3D models in vitro despite the presence of fibroblasts, thus further strengthening its potential viability and effectiveness against solid BC tumors in an in vivo setting.

The presence of IL-15 [31] and the removal of checkpoint inhibition have been identified as central elements in NK therapy of solid tumors [23]. We could demonstrate this functionality in our system and anticipate an enhanced function in patients. Similarly, the inclusion of a suicide gene is necessary to allow selective elimination of the cells.

In summary, we have developed novel CAR-modified target-specific NK cells as highly efficient allogenic immunotherapeutics against refractory and relapsed hematological and solid tumors.

## 4. Materials and Methods

### 4.1. Cell Lines and Cell Culture

The Jurkat cell line and the breast cancer cell lines MCF7, HCC1937, MDA-MB-231 and MDA-MB-468 were maintained in Roswell Park Memorial Institute (RPMI) 1640 medium (Gibco, NY, USA) supplemented with 10% inactivated fetal bovine serum (FBS; Gibco), 200 mM L-Glutamine and 1% penicillin/streptomycin (Pen/Strep; Gibco). Human embryonic kidney (HEK) 293T cells were cultured in Dulbecco’s modified Eagle’s medium (DMEM; Gibco) supplemented with 10% FBS, 200 mM L-glutamine and 1% penicillin/streptomycin (Pen/Strep; Gibco). YT cell lines were cultured in RPMI-1640 supplemented with 10% FBS, 10 U/mL IL-2 (Immunotools, Friesoythe, Germany), 200 mM L-glutamine and 1% penicillin/streptomycin (Pen/Strep; Gibco). Primary NK cells were isolated from buffy coats (DRK, German Red Cross; Dresden, Germany) using density gradient centrifugation followed by CD3 depletion (Stemcell Technologies, Koln, Germany). The isolated NKs were further cultivated in NK MACS medium (Miltenyi, Teterow, Germany), 5% AB serum (Merck, Darmstadt, Germany), 500 U/mL IL-2 (Immunotools), 140 U/mL IL-15 (Immunotools). All cells were cultivated in a humidified cell culture incubator at 37 °C, 5% CO_2_ and regularly tested for mycoplasma (MycoAlert Lonza, Koln, Germany). Required cells were stimulated with phytohaemagglutinin -l (PHA; Sigma-Aldrich, Darmstadt, Germany).

Breast cancer cell lines expressing nano-luciferase were generated by transducing the parental cell line with a dual reporter plasmid pCDH-EF1-Nluc-P2A-copGFP-T2A-Puro which was a gift from Kazuhiro Oka (Addgene plasmid #73022; http://n2t.net/addgene:73022 URL (accessed on 18 May 2023); RRID: Addgene_73022) and were maintained in culture under puromycin selection.

### 4.2. Lentiviral Vector Production and Transduction

Virus particle production was performed through transient transfection of HEK-293T cells using linear polyethylenimine (PEI). For each µg of plasmid DNA 100 µL of DMEM was used. All plasmid DNA was thawed and subsequently mixed into a reaction tube containing DMEM at a ratio of 5:4:3:1 (CAR: pMDLg: pRSV-Rev: pMD2G). The PEI stock solution (100 µg/µL) was diluted at a 1:13 (DNA: PEI) ratio in DMEM. The diluted PEI solution was then carefully added to the DNA solution and immediately pulse vortexed before it was left to incubate at RT for 20 min. Medium was aspirated from the preplated HEK cells, and DMEM was added to the cells followed by the dropwise administration of the transfection reagent. The cells were then left to preincubate at 37 °C, 5% CO_2_ for 3–4 h. After preincubation complete medium (DMEM + 10% FBS) was added to the plate and subsequently left for an overnight incubation at 37 °C, 5% CO_2_. The following morning, the medium was exchanged with complete medium. Viral supernatant was collected 12, 24 and 48 h after transfection and were subsequently centrifuged at 4 °C, 300× *g* for 10 min, followed by filtering it through a 0.22 µm syringe filter. The filtered supernatant was then ultra-centrifuged at 4 °C 25,000× *g* for 90 min and the viral pellet was resuspended in PBS. A small portion of the virus was aspirated and used for the viral titration, while the rest was aliquoted into the appropriate volumes, flash frozen, and stored at −80 °C.

For transduction, the target cells were resuspended in the appropriate media containing the produced viral particles and polybrene (5 µg/mL). Following resuspension, the cell suspension was left to react on a spinning rotator at RT for 60 min with a subsequent spin inoculation step at RT, 800× *g* for 120 min before the cells were left to incubate overnight in standard cell culture conditions. The next day, the transduction medium was replaced with the appropriate medium.

### 4.3. Cytotoxicity Assays

The Nano-Glo© luciferase assay system (Promega, Walldorf, Germany) was used to measure cytotoxicity following the manufacturer’s protocol. Briefly, cells were incubated at the indicated E:T ratios, treatments, and time points. Following incubation, an aliquot of cell culture medium was aspirated and mixed 1:1 with luminescence reaction reagent and subsequently measured using a Tristar 3 multimode plate reader (Berthold Technologies, Bad Wildbad, Germany). Specific target cell lysis was calculated using the released nanoluciferase from target cells alone and Triton-X lysed cells using the formula:$$\mathrm{Cytotoxicity}\;(\%)=\frac{Sample\;release-Low\;release}{High\;release-Low\;release}\times 100$$

### 4.4. Multicellular Tumor Spheroid Culture (3D Cell Culture)

Three-dimensional cell culture was performed using a liquid overlay technique in which 2 × 10^3^–5 × 10^3^ cells, depending on cell line and application, were seeded into a 96-well plate pre-coated with anti-adherence solution (Stemcell Technologies) and centrifuged at 200× *g* for 5 min. The plates were incubated in a humidified cell culture incubator at 37 °C, 5% CO_2_ for at least 48 h to allow tumor spheroid formation before further use. Cell lines used for the spheroid cultivation included TNBC cell line HCC1937 and the fibroblast cell line MRC5. For co-culture spheroids a 1:1 ratio of HCC1937 and MRC5 was used. Spheroid formation and further cultivation was performed in RPMI-1640 supplemented with 10% FBS (Gibco), 200 mM L-glutamine and 1% pen/strep (Gibco).

### 4.5. Flow Cytometry

For flow cytometry analysis, the collected cells were pelleted and washed once with PBS and then re-suspended with the appropriate amount of antibody and left to react for 30 min at 4 °C. Following antibody incubation, the cells were washed twice with PBS for 5 min at 300× *g* before they were resuspended in PBS with 0.3% formaldehyde and measured in the flow cytometer. Antibodies used were CD3 (clone HIT3b), CD16 (clone 3G8), CD56 (clone BA19), CD57 (clone HI57), CD62L (clone HI62L), CD279 (clone EH12.2H7) and NKG2D (clone 1D11) supplied by Immunotools. Anti-CD94 (clone DXX22), CD274 (clone 29E.2A2) streptavidin PE and AF647, and IgG Fc (clone QA19A42) were supplied by Biolegend. Anti CD44v6 (clone 2F10) was supplied by R&D Systems, Minneapolis, MN, USA. Polyclonal anti-IgG Fc was supplied by Jackson Immunoresearch, West Grove, PA, USA. Fluorescence was measured on a BD FACScalibur™ and the data was analyzed with FlowJo v10.8 (Becton Dickinson and Company, Franklin Lakes, NJ, USA; 2019).

### 4.6. Immunocytochemistry and Confocal Microscopy

MCTS were collected and washed three times in PBS by allowing the spheroids to settle in the bottom of the reaction tube before aspirating the supernatant, the washing step was repeated three times in between every treatment step. Following the initial washing steps, the MCTS were fixed in 4% paraformaldehyde for 20 min and subsequently permeabilized with 0.1% Triton X-100 for 20 min, blocked with 3% BSA and Beriglobin^®^ 1 mg/mL for 1 h and then left to react over night at 4 °C with the indicated antibody. After antibody incubation, the spheroids were mounted on a microscope slide using ROTI^®^Mount FluorCare mounting solution. The confocal imaging was acquired with a Leica TCS SPE confocal microscope and the image analysis was performed using Image J 1.54d (U.S. National Institutes of Health, Bethesda, MD, USA).

### 4.7. Statistical Analysis

Statistical analyses were conducted using Prism (v. 8.4.2, Graphpad Prism, San Diego, USA). Normal distribution was confirmed, and assay data were compared using student’s *t*-test or ANOVA. Statistical significances are marked *p* < 0.05 = *, *p* < 0.01 = **, *p* < 0.001 = *** and nonsignificance is marked with ns.

## Figures and Tables

**Figure 1 ijms-24-09038-f001:**
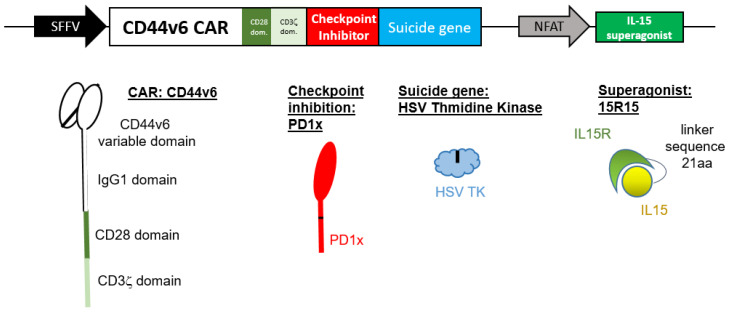
Design of a novel CD44v6 CAR vector. A novel lentiviral CAR vector specific for CD44v6 was designed. The vector includes checkpoint inhibition and an IL-15 superagonist expressed under the control of an NFAT-responsive promoter. In order to be able to eliminate CAR-expressing cells, we have included a suicide gene (HSV thymidine kinase; HSV TK). CD44v6 CAR, checkpoint inhibition and HSV TK are under the control of the constitutive SFFV promoter and linked by 2A sequences.

**Figure 2 ijms-24-09038-f002:**
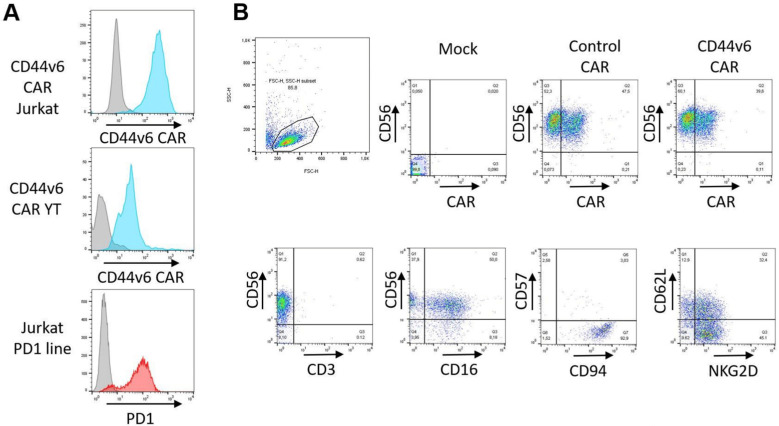
CD44v6 CAR expression on cell lines and primary NK cells. The expression of CD44v6 CAR on transduced cells was determined by flow cytometry. (**A**) Transduced Jurkat reporter cells and YT cell lines expressed high levels of the CD44v6 CAR (Blue curves). The original Jurkat PD1 line also expressed high levels of PD1 (Red curve). (**B**) Primary NK cells transduced with CAR constructs were stained either for markers relevant to NK function (lower panel) or expression of CD44v6 or control CAR where the recognition domain is missing (upper right panel).

**Figure 3 ijms-24-09038-f003:**
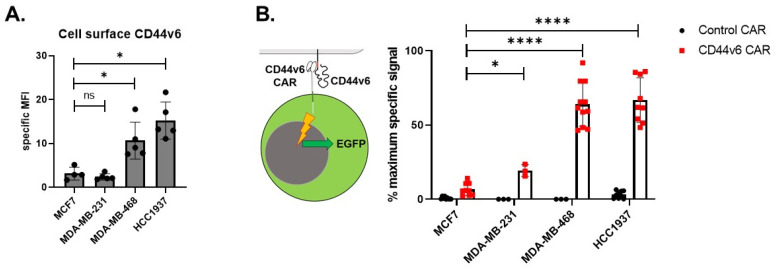
Novel CD44v6 CAR efficiently recognizes CD44v6 expressed on breast cancer cell lines. Flow cytometry was used to test the level of CD44v6 expressed in breast cancer cell lines and the response of Jurkat reporter cells. (**A**) Flow cytometric evaluation of the surface expression of CD44v6 on 1 receptor-positive (MCF7) or 3 triple negative breast cell lines is shown, *n* = 5–10 biological replicates. Results are given as mean fluorescence intensity (MFI). Two-tailed unpaired *t*-test with Mann–Whitney correction was used to determine significance. Error bars represent the mean ± SD; * *p* < 0.05. (**B**) In order to validate our construct, we used Jurkat reporter cells encoding EGFP under the control of an NFkB promoter. These cells were transduced with CD44v6 CAR or a control construct lacking the CD44v6 recognition domain. Jurkat reporter cells were added to breast cancer cell lines on 96 well plates for 18 h at an E:T ratio of 1:1. The increase in GFP and Jurkat reporter cells upon exposure to breast cancer cells or stimuli was taken as specific CAR stimulation. Assay is depicted graphically in the left section of the panel. Results are shown as % of maximum stimulus (PHA 1 μg/mL) *n* = 3 biological replicates done in triplicate. **** *p* < 0.0001; * *p* < 0.05; ns not significant.

**Figure 4 ijms-24-09038-f004:**
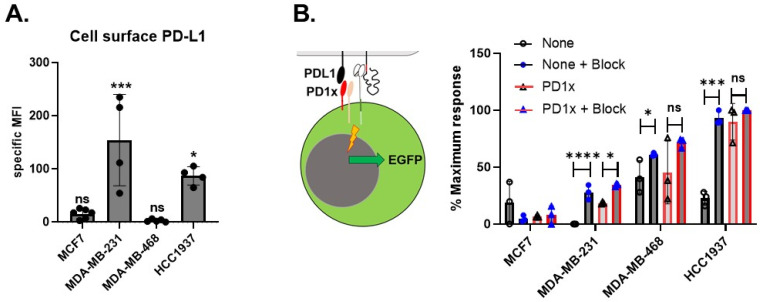
Checkpoint molecules do not inhibit novel CD44v6 CAR. In order to test the level of PD-L1 expressed on breast cancer cell lines and its effect on Jurkat reporter cells, we used flow cytometry. (**A**) Surface expression of PD-L1 on 1 receptor-positive (MCF7) or 3 triple negative breast cell lines is shown, *n* = 4–6 biological replicates. Results are given as mean fluorescence intensity (MFI). Two-tailed unpaired *t*-test with Mann–Whitney correction was used to determine significance. Error bars represent the mean ± SD. (**B**) In order to test checkpoint inhibition, we used reporter Jurkat cells expressing high levels of PD1, making them highly sensitive to checkpoint inhibition. These cells were then transfected with CD44v6 CAR (PD1x) or a CD44v6 CAR without checkpoint inhibition (none). After mock treatment or treatment with 10 μg/mL of pembrolizumab and atezolizumab for 1 h (block), cells were then incubated at a ratio of 1:1 with breast cancer cell lines for 24 h in 96-well plates. The increase in GFP and Jurkat cells, as determined by flow cytometry upon exposure to cells or stimuli, was taken as specific CAR stimulation. Assay is depicted graphically in the left section of the panel. Results are shown as % of maximum stimulus (PHA 1 μg/mL) *n* = 3 biological replicates done in triplicate. In the absence of vector-encoded checkpoint inhibition, breast cell lines with strong expression of PD-L1, such as HCC1937, showed low CD44v6-specific activation of Jurkat reporter cells. This was corrected either by vector-encoded checkpoint inhibitor (PD1x) or by the addition of pembrolizumab and atezolizumab (block), indicating it was due to checkpoint signaling via PD1. **** *p* < 0.0001; *** *p* < 0.001; * *p* < 0.05; ns not significant.

**Figure 5 ijms-24-09038-f005:**
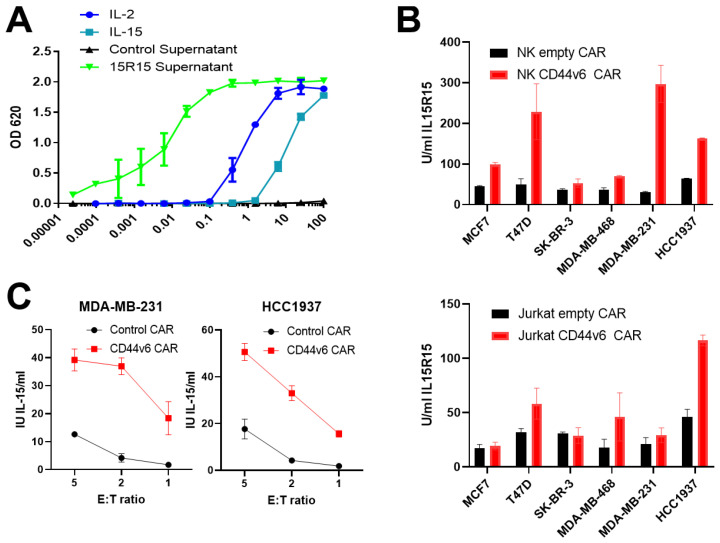
CD44v6 CAR stimulation induces production of IL-15 superagonist from NFAT promoter. In order to maximize stimulation of NK cells, we designed an IL-15 superagonist (15R15) controlled by an NFAT promoter. (**A**) In order to test the 15R15 molecule, we took supernatant from Hek293T cells transiently transfected with a plasmid encoding 15R15 under the control of a constitutive promoter. Addition of the supernatant to an IL-2/IL-15 responsive cell line showed strong stimulation (green line) equivalent to or superior to recombinant IL-2 or IL-15 (blue and turquoise lines). (**B**) Jurkat cells (upper panel) or primary NK cells (lower panel) were transduced with CD44v6 or a control construct lacking the CD44v6 recognition domain. Effector cells were added at a ratio of 1:1 to monolayers of target cells in 96-well plates. Supernatant was collected after 72 h and tested for IL-15 activity by bioassay. (**C**) Jurkat cells, as for B, were added to HCC1937 cells at different effector-target ratios. Supernatant was collected after 24 h and tested for IL-15 activity by bioassay. *n* = 3 biological replicates were performed in duplicate.

**Figure 6 ijms-24-09038-f006:**
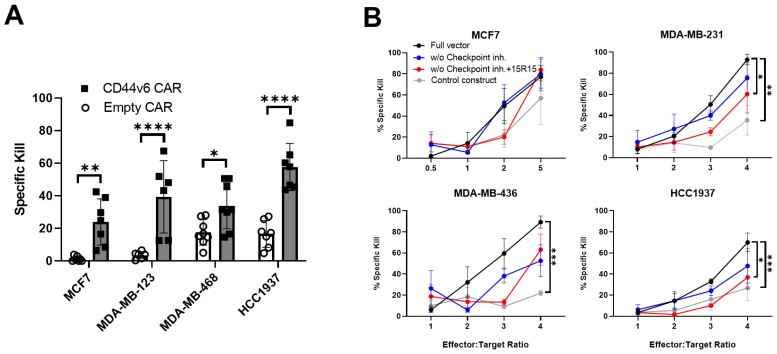
CD44v6 CAR kills breast cancer cell lines in vitro. In order to test the functionality of the CD44v6 CAR, we transduced cell lines and primary NK cells with CD44v6 CAR vector. (**A**) YT cells transduced with CD44v6 CAR (black) or a control construct lacking the CD44v6 recognition domain (empty) were tested for cytotoxicity to BC. Effector cells were added at a ratio of 5:1 to monolayers of target cells expressing cell-associated nanoluciferase in 96-well plates and incubated for 18–24 h before supernatants were collected and assessed for luciferase, indicating cell lysis or kill. Results are shown as % of maximum kill (cellular lysis; triton X-100 1% in PBS) *n* = 3 biological replicates were performed in duplicate. (**B**) Primary human NK cells transduced with the full CD44v6 CAR vector (black) or with variants lacking checkpoint inhibition (blue), both checkpoint inhibition and IL-15 production (red) or control CAR vector (grey) were tested for cytotoxicity to 4 breast cancer cell lines, including 3 triple negatives. Effector cells were added at different ratios to monolayers of target cells expressing cell-associated nanoluciferase in 96-well plates and incubated for 8 h before supernatants were collected and assessed for luciferase, indicating cell lysis or kill. Results are shown as % of maximum kill (cellular lysis; triton X-100 1% in PBS) *n* = 2–4 biological replicates were performed in duplicate. Statistical significance was determined with two-way ANOVA with Tukey’s multiple comparisons test. **** *p* < 0.0001; *** *p* < 0.001; ** *p* < 0.01; * *p* < 0.05.

**Figure 7 ijms-24-09038-f007:**
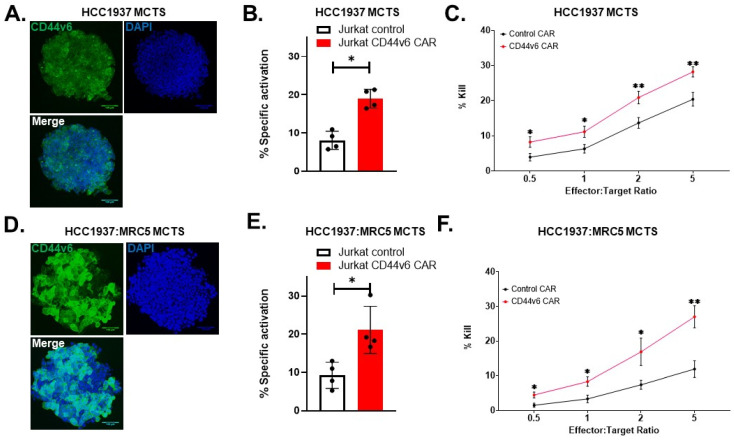
CD44v6 CAR reacts to TNBC MCTS and primary natural killer cells show cytotoxic potential to TNBC tumor spheroid models. CD44v6 CAR-NK cytotoxicity in triple-negative breast tumor spheroid models. Primary NK cells expressing CD44v6 CAR were tested on 3D triple-negative breast cancer tumor spheroid models. The addition of fibroblasts augments the MCTS to better model the tumor microenvironment and cancer-associated fibroblasts to its in vivo counterparts. (**A**,**D**) The high levels of CD44v6 expression on HCC1397 cells are unaffected or enhanced by the presence of fibroblasts (MRC5), as shown by confocal images of tumor spheroids stained for CD44v6 (green) and counterstained with DAPI (blue). Spheroids had a size of approximately 500 µm and were imaged in a confocal microscope at ×20 magnification. (**B**,**E**) MCTS expressing CD44v6 were recognized by the CD44v6 CAR, as shown by the stimulation of reporter cells. Jurkat reporter cells were added at a 1:1 ratio with HCC1937 or HCC1937:MRC5 tumor spheroid for 18 h in a 96-well plate. Specific activation of Jurkat reporter cells was calculated by measuring GFP expression of the Jurkat population by flow cytometry and results are shown as % specific activation; *n* = 4 biological replicates were performed in triplicate. Statistical significance was determined by *t*-test. (**C**,**F**) Primary CD44v6 CAR-NK cells were tested for cytotoxicity against tumor spheroid models. Primary human NK cells transduced with CD44v6 CAR or an identical vector lacking a recognition domain on the CAR (control CAR) were added at the indicated E:T ratios to tumor spheroids. Cells were incubated for 18 h before supernatants were collected and measured for released luciferase, indicating cell lysis or kill. Results are shown as percentage of maximum lysis, (triton X-100 10% in PBS); *n* = 4 biological replicates done in triplicate (**C**), *n* = 2 biological replicates done in triplicate (**F**). Statistical significance was determined using a multiple *t*-test with Holm–Sidak correction. Error bars represent the standard error of the mean ± SEM. ** *p* < 0.01; * *p* < 0.05.

## Data Availability

Not applicable.

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
