# Peer review of "Next Generation CD44v6-Specific CAR-NK Cells Effective against Triple Negative Breast Cancer"

_ijms, 2023, doi:10.3390/ijms24109038_

Round 1

Reviewer 1 Report

The authors generated Chimeric Antigen Receptor (CAR) Natural Killer (NK) cells using CD44V6 and IL-15  with competitive Inhibitors of PDL-1. This CAR-NK CD44v6 showed the improved cytotoxic killing of TNBC cells (MDA-MB-231, MDA-MB-468, HCC1937) and 3D spheroids with an improved specific release of IL-15 superagonist. The work provides an approach to improve cytotoxic killing in TNBC tumors not based on CAR-T cells which need to be autologous, but based on NK cells which will provide better results and availability.

The manuscript overall is well organized and easy to follow with relevant experimental design.

Comments

General comments:

If the authors claim the specificity of this CAR-NK CD44v6 to TNBC, other non-TNBC cells could be tested to solidify the conclusion.

It is interesting to know the expression profile of other CD44 variants in such cells and if these cells express other variants at higher levels than CD44v6. As variants are very similar, and the majority of differences are in the intracellular domains.

Figure 3A: MCF7 is a non TNBC cell line, yet it shows higher expression than MDA-MB-231. However, in Figure 3B, the CD44V6 CAR recognized CD44v6 on MDA-MB-231 better than MCF7(any explanation) or the possibility of cross-recognition with other CD44 variants?

In line 149 the author stated, “Overall, this data indicates that our next generation CD44v6 CAR construct shows specific activity against TNBC lines.” I think in order to make such a statement, the author needs to test several non-TNBC cell lines and demonstrate that the CAR-NK is specific only to TNBC cells, particularly the author using MCF-7, which is non-TNBC and MCF-7 show response.

Figure 6 has no statistical significance asterisks or statistical analysis described in the results or legends.

Thank you

Author Response

Reviewer 1: If the authors claim the specificity of this CAR-NK CD44v6 to TNBC, other non-TNBC cells could be tested to solidify the conclusion:

Reply: Our intent was to show that TNBC was targeted by the CD44v6 CAR at least as well as non-TNBC, thus making this an effective therapy for TNBC where there are few therapy options available. Several other BC lines (e.g. T47D or SKBR3) express CD44v6 and can be attacked by the CD44v6 CAR.

Reviewer 1: It is interesting to know the expression profile of other CD44 variants in such cells and if these cells express other variants at higher levels than CD44v6. As variants are very similar, and the majority of differences are in the intracellular domains.

Reply: To the best of our knowledge no antibody panel exists that can reliably detect all variants of the highly-spliced CD44 molecule. qRT-PCR data of the transcriptome would be informative and the topic is certainly interesting, particularly in light of the recent publication of Solier (nature 2023) with regard to CD44. We agree with the reviewer that other variants are probably better expressed than CD44v6, however how the variants interact is still insufficiently understood and is beyond the scope of this paper.

Reviewer 1: Figure 3A: MCF7 is a non TNBC cell line, yet it shows higher expression than MDA-MB-231. However, in Figure 3B, the CD44V6 CAR recognized CD44v6 on MDA-MB-231 better than MCF7(any explanation) or the possibility of cross-recognition with other CD44 variants?

Reply: We agree with the reviewer that the activity of the CD44v6 CAR does not map 1:1 with the expression of the antigen on target cells. However, we have not seen the CD44v6 CAR specifically activate in the absence of CD44v6. It is more probable that factors present on particular BC lines (e.g. CD44 variants) cause steric blockage of the antigen. Alternatively, adhesion factors, costimulation/inhibition factors or cytokines expressed by specific tumor cell lines could interfere with cytotoxic cell function to a greater or lesser extent.

Reviewer 1: In line 149 the author stated, “Overall, this data indicates that our next generation CD44v6 CAR construct shows specific activity against TNBC lines.” I think in order to make such a statement, the author needs to test several non-TNBC cell lines and demonstrate that the CAR-NK is specific only to TNBC cells, particularly the author using MCF-7, which is non-TNBC and MCF-7 show response.

Reply: Our intent was to show that TNBC was targeted by the CD44v6 CAR at least as well as non-TNBC. The CD44v6 construct shows specific activity against both TN and non-TNBC, and we have now modified our paper to allow for activity against non-TNBC. The line now reads” Overall, this data indicates that our next generation CD44v6 CAR construct shows specific activity against BC lines, including TNBC.”

Reviewer 1: Figure 6 has no statistical significance asterisks or statistical analysis described in the results or legends.

Reply: We have now included statistical analysis and simplified panel A to improve this figure.

Reviewer 2 Report

General: The submitted manuscript presents the new-generated CAR targeting CD44v6 aiming to offer a new therapeutic option for the treatment of triple-negative breast cancer. The CAR includes “CD44v6-63 specific Vl and Vh recognition domains with the hinge region of IgG1, the transmembrane 64 and signaling domains of CD28 and the signaling domain of CD3ζ. This was coupled by P2A sequences to a checkpoint inhibitor (PD1x) and a suicide gene (Thymidine 66 kinase from herpes simplex virus type 1; HSV TK). This was followed by a response element with an IL-15 superagonist (15R15) under the control of a NFAT promoter”.

This is very careful made study dedicated to identification the tumor toxic effect in vitro of novel CAR-modified target-specific NK cells both in 2D and 3D cellular systems.

Methods used are proper, results are good founded, and conclusions are correct. 

In additional, very good illustrative materials are presented.

Minor:

1.      Please provide the numeric/graphic data of CD94 and NKG2D expression (lines 85-95).

2.      Figure 5 is hard for examination due to its page position. Please move it into a more acceptable position.

Conclusion:

This study merits publication and would be of interest to the readership of the journal as well as other researchers at medical and biological sciences. Manuscript should be accepted and published after the minor corrections.

Author Response

Reviewer 2: Please provide the numeric/graphic data of CD94 and NKG2D expression (lines 85-95).

Reply: This is shown in panel B of Figure 2, and we have amended the text to clarify this point.

Reviewer 2: "Figure 5 is hard for examination due to its page position. Please move it into a more acceptable position.

Reply: We have now adjusted the placement of this figure.